# In Silico and In Vivo Evaluation of Synthesized SCP-2 Inhibiting Compounds on Life Table Parameters of *Helicoverpa armigera* (Hübner)

**DOI:** 10.3390/insects13121169

**Published:** 2022-12-16

**Authors:** Qamar Saeed, Faheem Ahmad, Numan Yousaf, Haider Ali, Syed Azhar Ali Shah Tirmazi, Abdulrahman Alshammari, Naeema Kausar, Mahmood Ahmed, Muhammad Imran, Muhammad Jamshed, Metab Alharbi, Muhammad Muddassar

**Affiliations:** 1Department of Entomology, Faculty of Agricultural Sciences and Technology, Bahauddin Zakariya University, Multan 60800, Pakistan; 2Department of Biosciences, COMSATS University Islamabad, Park Road, Islamabad 45550, Pakistan; 3School of Chemistry, University of the Punjab, Lahore 54590, Pakistan; 4Department of Pharmacology and Toxicology, College of Pharmacy, King Saud University, P.O. Box 2455, Ryadh 11451, Saudi Arabia; 5Department of Chemistry, Division of Science and Technology, University of Education, College Road, Lahore 54000, Pakistan; 6KAM-School of Life Sciences, FC College (A Chartered University), Lahore 54000, Pakistan; 7Department of Biological Sciences, University of Calgary, 2500 University Dr. NW, Calgary, AB T2N 1N4, Canada

**Keywords:** synthetic compounds, molecular docking, MD simulations, life table, SCP-2 inhibitor

## Abstract

**Simple Summary:**

Different crops and vegetables are attacked by a complex of pest insects causing severe losses in their yield and quality, among them *Helicoverpa armigera* is one of the key pests. Its polyphagous status and extended environmental tolerance, coupled with high fertility, fecundity rate and short generation time, enables it to quickly attain a primary pest status in any suitable host. To minimize losses caused by this notorious pest, multiple applications of chemical pesticides are required. Due to such indiscriminate use of synthetic insecticides, environmental and human health hazards have become a major concern of the modern-day plant protection industry. Novel scaffolds that may regulate insect growth can offer a sustainable alternative to conventional insecticides. In this study, synthesized small molecules, with a tendency to disrupt insect molting, were evaluated against a *Helicoverpa armigera*. One of the tested compounds significantly reduced larval and pupal weight accumulations and prolonged stadia lengths resulting in disrupted population growth. At the same time, the emerged females had significantly reduced fertility. These findings suggest that further optimization of tested scaffold may lead to help finding new insecticide-like molecules that will reduce the dependence on traditional chemical insecticides.

**Abstract:**

For environment-friendly, safe and nonpersistent chemical control of a significant polyphagous insect pest, *Helicoverpa armigera*, discovery of growth-regulating xenobiotics can offer a sustainable alternative to conventional insecticides. For this purpose, chemically synthesized compounds to inhibit sterol carrier protein (SCP-2) function using in silico and in vivo assays were evaluated to estimate their impact on the survivals and lifetable indices of *H. armigera*. From nine chemically synthesized compounds, OA-02, OA-06 and OA-09 were selected for this study based on binding poses mimicking cholesterol, a natural substrate of sterol carrier protein and molecular dynamics simulations. In vivo bioassays revealed that all compounds significantly reduced the larval and pupal weight accumulations and stadia lengths. Subsequently, the pupal periods were prolonged upon treatment with higher doses of the selected compounds. Moreover, OA-09 significantly reduced pupation and adult emergence rates as well as the fertility of female moths; however, fecundity remained unaffected, in general. The life table parameters of *H. armigera* were significantly reduced when treated with OA-09 at higher doses. The population treated with 450 μM of OA-09 had the least net reproductive rates (Ro) and gross reproductive rate (GRR) compared to the control population. The same compound resulted in a declining survival during the early stages of development coupled with reduced larval and pupal durations, and fertility. These results have a significant implication for developing an effective and sustainable chemical treatment against *H. armigera* infestation.

## 1. Introduction

Insecticides have always been a main component of conventional and integrated pest management practices [1]. In the high-income countries, the application of pesticides has helped to increase crop development and yield significantly by reducing losses caused by noxious insect pests. However, intensification of agriculture with excessive use of pesticides has resulted in serious health and environment issues and among them environmental pollution and food contaminations are at the top [2]. During the 1950s, use of pesticides in agricultural landscapes increased exponentially due to plant diseases and cultivation of new land along with the need to boost yields to cater for the world’s increasing dietary requirements [3,4]. The United States alone imports more than 350 MT of pesticides, whereas the import of chemical pesticides has touched the figure of 250 MT per annum in recent years [5,6]. The world population approaching 11 billion demands an increase in quality and quantity of available agricultural commodities, and if such pattern of indiscriminate use of insecticides persists, environmental impact will increase many folds in no time [7,8]. A multitude of different formulations are currently available and consumption of more than 65 MT of active pesticide ingredients in the world [9] strongly mobilize the agricultural communities and environmental activists to struggle for the use of safer alternatives.

Cotton is treated as one of the main cash crops [10] in subtropical and tropical parts of Asia, Africa and the Americas [11,12,13]. One of the major factors in crop yield decline can be attributed to insect losses [14,15]. During the whole cropping season, cotton crop is attacked by a complex of pest insects causing severe losses in yield and quality and *Helicoverpa armigera* (Hübner) (Lepidoptera; Noctuidae) is one of the key pests that require multiple applications of chemical pesticides [16]. Its polyphagous status [17] and extended environmental tolerance, coupled with high fertility and fecundity rates and short generation times, enables it to quickly attain a primary pest status in any suitable host [18,19]. Being a polyphagous pest, *H. armigera* is reported to cause severe losses to cotton, corn and soybean in US agroecosystems [20,21], tomato and tobacco in Brazil [22,23] and also soybean, sorghum, maize, vegetables and citrus in south American regions [17,24,25]. It has more than 180 wild and cultivated host species worldwide with at least forty cultivated families including Solanaceae, Poaceae, Malvaceae, Asteraceae and Fabaceae [26,27]. Since the larvae primarily feed on flowers and fruit, a direct yield loss is observed [28,29,30,31,32].

Farmers, in general, invest more to protect cash crops and the same is the case here in Pakistan, where according to an estimate, more than 80% of the pesticides that are imported to the country are applied in cotton agroecosystem [33]. Although this usage has increased cotton yields three-fold, it also poses a serious threat of environmental contamination and health hazards [34]. Due to excessive and indiscriminate use of chemical pesticides, *H. armigera* has developed strong resistance against organophosphates, organochlorines, carbamates and pyrethroids and even to *Bt*-gene Cry-1 in transgenic cultivars of cotton [35,36,37,38,39,40]. The farmers, therefore, have now started using parallel management practices for its control by managing crop rotation and cultivation adjustments [41], using resistant varieties, installing pheromone traps, shifting to genetically modified varieties, removal of alternate hosts and using biological control agents [24,27]. However, discovery of a novel mode of action synthetic molecules that may target insect physiological pathways is often neglected. 

Use of synthetic and target specific compounds such as the insect growth regulators (IGR) could be one of the alternative and safe approaches to develop new pesticides. Such compounds provide effective and environmentally safe insecticides that are selective and nonpersistent with the least health risks for humans and animals [42,43,44]. Already available IGRs in the market belong to the family of chitin synthesis inhibitors such as hexaflumuron and chlorfluazuron [42], and insects’ juvenile hormone mimic, e.g., pyriproxyfen [45]. Further expansion in the field of identification of insect metabolic pathways will expand the options for pesticide discoveries. One such newly identified pathway, is the sterol production pathway [46]. Insects and other organisms use sterols as a structural and functional component of membrane rigidity and fluidity [16,47]. In this pathway, the sterols work as a precursor of molting hormones including 20-hydroxyecdysone, ecdysone and makisterone A. The most common form of a sterol is cholesterol that is often found in the mid-guts of the insects [28,48]. Like plants and mammals, insects are not able to produce sterol in the body, they fulfill the requirement of sterol from their food [49,50] and then use a sterol carrier protein-2 (SCP-2) to extract cholesterol. It is a nonspecific intracellular lipids carrier that is reported to play a vital role in the transport of cholesterol in humans, rats and insects [51]. The existence of sterol carrier proteins (SCP-2) has already been reported from major insect orders, including Diptera, Hymenoptera, Lepidoptera, Hemiptera and Coleoptera [52,53]. Blocking SCP-2 can result in delayed larval development, interrupted molting, deformities and reduced fecundity [54,55]. Moreover, some studies have reported the potential of SCP-2 inhibitor molecules to reduce the activities of insects’ chemical detoxifying enzymes, i.e., GST and P450, etc. [53,56].

Management of *H. armigera* has crucial ecological and environmental significance [57,58] as many of the potent synthetic insecticides are being reported from different regions to have reduced efficacy due to resistance development [54,59] in the pests. Identifying a novel mode of action compounds can deal with this serious concern. In this study, we have identified three quinolone azines synthetic compounds, i.e., OA-02 ((E)-7-((E)-benzylidenehydrazono)-10-(4-methylpiperazin-1-yl)-3,7-dihydro-2H-[1,4]oxazino[2,3,4-ij]quinoline-6-carboxylic acid), OA-06((E)-7-((E)-benzylidenehydrazono)-9-fluoro-3-methyl-10-(4-methylpiperazin-1-yl)-3,7-dihydro-2H [1,4]oxazino[2,3,4-ij]quinoline) and OA-09 ((E)-7-(hydrazono)-9-fluoro-3-methyl-10-(4-methylpiperazin-1-yl)-3,7-dihydro-2H-[1,4]oxazino[2,3,4-ij]quinoline-6-carboxylic acid) using in silico methods and then evaluated them in vivo against *H. armigera* under controlled conditions. The findings show that these synthetic compounds after optimization could have significant potential in controlling the polyphagous insect species.

## 2. Materials and Methods

### 2.1. Toxicity Predictions

The toxicity risks of the OA series compounds were predicted by DataWarrior [60]. The predicted properties contained tumorigenicity, reproductive effect, irritant effect and mutagenicity.

### 2.2. Molecular Docking

The NMR structure of sterol carrier protein (PDB ID: 4UEI) was prepared by the protein preparation wizard in Schrodinger Maestro [61]. The protein preparation involved assigning bond orders, addition of hydrogens, creating disulfide bonds and zero-order bonds to metals, removing water molecules from hetero groups beyond 5 Å and leaving the hetero state in its default pH (7.0). The hydrogen bonds were optimized by using PROPKA at pH 7.0. Finally, restrained energy minimization was performed using the OPLS_2005 forcefield [62]. After preparation of the receptor, a 3D grid was generated at particular sites to conduct site-specific docking. The values of internal coordinates for X, Y and Z were 7.67, −8.81 and −11.16, respectively. Similarly, the compounds were prepared using the LigPrep tool [63]. Various conformations of rings and stereoisomers of ligands were generated. The OPLS_2005 forcefield was used to optimize and minimize the 3D conformers of ligands. The prepared ligands were docked to the receptor grid by using the Glide docking module in SP (Standard Precision) mode [64]. The docked ligands were analyzed based on their binding modes and glide score. The compounds with plausible binding modes were selected for MD simulation to estimate their binding stability.

### 2.3. Molecular Dynamics Simulation

Three complexes were selected based on the binding modes. The selected complexes were explored to check binding stability by a 50 ns long MD simulation using the VMD [65] and NAMD tools [66]. The initial files were prepared through AMBER21 tools [67]. Antechamber was used to generate the ligand topology files, while the missing hydrogen were added to protein by the LeaP program [67]. In order to mimic the physiological environment, a 10 Å periodic boundary solvation box of TIP3P water molecules [68] was added to complexes. The solvated systems were neutralized by Na^+^ and Cl^−^ counter ions. The systems were minimized for 20 fs to avoid energy clashes. The equilibration of solvation systems was conducted at 310 K. To maintain the system stability, three additional equilibrations were conducted at 200 K, 250 K and 300 K, respectively. The stabilized systems were subjected to 50 ns simulation in the production run. The MD trajectories were stored at every 2 ps and analyzed by CPPTRAJ [69] and Bio3D package [70] of R program. Moreover, the binding free energies of the complexes were calculated by the MM/GBSA module of AMBER21 tools.

### 2.4. Larval Diet and Test Concentration Preparation

The selected compounds were assayed for their biological activity against the 3rd, 4th and 5th instars larvae of *H. armigera* using a diet incorporation method [71]. Artificial larval diet was prepared using the methods and ingredients described by Hamed and Nadeem (2008). Three concentrations of the test compounds, i.e., 50 µM, 150 µM and 450 µM were mixed at the rate of 40 µL/2 g of prepared diet.

### 2.5. Collection and Rearing of Cultures

Freshly laid egg masses and young larvae of *H. armigera* were collected from the cotton crop cultivated in the research fields of the Cotton Research Institute (CRI), Central Cotton Research Institute (CCRI) and research fields of the Bahauddin Zakariya University, Multan, to start the laboratory cultures for bioassays. The field collected larvae were kept individually to confine any infections they might have and minimize cannibalism [72] and were reared on freshly collected cotton leaves until they pupated. From the next generation, the larvae were fed on an artificial diet for two generations to exclude any effect of field diets. Glass containers (500 mL) covered with muslin clothes were used to house the larvae and the cultures were kept at 25 ± 2 °C and 75 ± 5% r.h. with a 14:10 h light:dark regime. To facilitate pupation, after the 4th molt, the larvae were supplied with sterile soil at the base of each jar as a medium for pupation. Later, the pupae were retrieved from the soil media and were transferred to clear jars (5L) until adult emergence. The moths, when eclosed, were provided with 10% honey solution [73] and strips of viscose tissue, hanging vertically as oviposition surfaces. Such rearing on artificial diet was continued until the second filial generation and then the larvae were exposed to the test compounds in bioassays. The master cultures in the laboratory population were augmented regularly by introducing newly collected field populations, when available.

### 2.6. Life Table Bioassay

The life table parameters of *H. armigera* were studied using the test concentrations mixed with artificial diet, while the plain diet (with no tested compound mixed) was used as a control. The experiment was replicated three times with 20 larvae of each instar. After weighting and recording the size of each larva, 2 g diet/larva were provided in each Petri dish containing a single larva. To avoid any fungal infections in the larval colonies, the diet was replaced on alternate days until pupation. The larvae health and survival were observed daily, and the stadia length was recorded. Three days after pupation, the pupae were weighed and kept in sterilized jars until adult eclosion, and their pupal periods were recorded. The newly emerged moths were sex sorted based on color of the forewings (males have greenish while the females have brownish forewings) [74]. Five pairs of adults from each treatment were transferred into oviposition jars with a supply of 10% honey solution and vertically hanging viscose tissue strips, as described earlier. The data for fecundity and fertility were recorded daily, and the pre-oviposition periods and adult longevity was noticed at the end of oviposition and until the mortality of the female moths, respectively.

Age-stage, two sex life table analyses [75] using fecundity, pre-oviposition period, total pre-oviposition period, larvae, adult and eggs were conducted following the model described by [76,77]. Survival (*l*_x_) and fecundity (*m*_x_) rates were calculated from proportion of larvae survived and the number of eggs laid per female followed by calculation of population growth parameters, such as net reproductive rate (*R*_o_), intrinsic rate of population increase (*r*), finite rate of population increase (λ) and generation time (*T*), using the equations given below:(1)$$l_{x}=\sum_{j=1}^{k}S_{xj}$$
(2)$$m_{x}=\overset{k}{\underset{j=1}{\sum }}S_{xj}{ʄ}_{xj}/\overset{k}{\underset{j=1}{\sum }}S_{xj}$$
(3)$$R_{o}=\overset{\infty }{\underset{x=0}{\sum }}\;l_{x}m_{x}$$

The population parameter intrinsic rate “r” was calculated to follow the iterative bisection methodology of Euler–Lotka formula:(4)$$\overset{\infty }{\underset{x=0}{\sum }}e^{-r\left(x+1\right)}l_{x}m_{x}=1$$

Mean generation time was calculated as:*T* = In*R_o_*/*r*
(5)
where “*k*” = number of stages, “*S_xj_*” = survival rate at specific stage (*j*) and age in days (*x*), “*ʄ_xj_*” = age-stage specific female fecundity, “*l_x_*” = age specific survival rate, “*m_x_*” = age specific fecundity, “*R_o_*” = net reproductive rate and “*T*” = mean generation time. The age index ranged from 0 to ∞ and finite rate “*λ*” was considered as $e^{r}$. (Goodman, 1982).

At the end, variance and standard error of fecundity, developmental time, adult longevity and other life table characteristics were calculated by bootstrapping [78] and the data were conformed to the assumption of equal variance. The means were analyzed with ANOVA followed by post hoc pairwise comparisons using Tukey’s HSD tests [76].

### 2.7. Statistical Analyses

The data regarding developmental parameters (larval, pupal, and adult longevities, pupation, and adult emergence) and reproductive potentials (fecundity and fertility) were analyzed using a 2-way factorial ANOVA using the compounds and their concentrations as independent factors, followed by Tukey’s HSD post-hoc pairwise comparisons. Since the data regarding larval mortality did not conform to the assumption of homogeneity even after appropriate data transformation (arcsine square root transformation) was applied, we analyzed the effects of compound type and their concentrations on larval mortality individually, using a nonparametric Kruskal–Wallis H test. The LC_50_ and LC_90_ values were calculated by fitting the probit model. All the above-mentioned analyses were conducted using SPSS software.

## 3. Results

### 3.1. Toxicity Predictions

The toxicity properties including tumorigenicity, reproductive effect, irritant effect and mutagenicity were predicted by DataWarrior. As shown in Table 1, OA-01 showed high mutagenicity, irritant effect, and reproductive effect. Other compounds did not show high toxicity properties.

### 3.2. Molecular Docking

The compounds were docked to the binding sites of SCP-2 protein to find their binding modes. All the compounds were ranked based on the glide score (Table 2). The docked compounds did not reside in the same binding pockets. The interactions of all compounds with the SCP-2 are shown in Appendix A Appendix A. The docked ligands were further analyzed based on the binding poses and three compounds, i.e., OA-02, OA-06 and OA-09 were selected as those had the same binding pose as the natural substrate cholesterol. The hydrophobic moieties of selected compounds interacted with the same residues and reside in the same position in the binding pocket (Figure 1A–F).

### 3.3. MD Simulation

The binding modes of selected compounds were further validated by 50 ns MD simulation. The MD trajectories were analyzed to calculate the root mean-square deviation (RMSD), root mean-square fluctuations (RMSF) and solvent accessible surface Area (SASA).

The RMSD of backbones atoms of apo protein and its complexes was calculated to estimate the stability of the protein. It can be observed that all systems were equilibrated at 5 ns. After equilibration, the RMSD of apo protein (blue) remained in the range of ~3.5–4 Å. A major deviation was observed during 30 to 40 ns where RMSD increased to ~6 Å but it attained the previous conformation at 40 ns and then remained in the same range until the end of simulation (Figure 2A). The RMSD plots of SCP-2 complexes were compared to apo protein. The major deviations in the RMSD were observed in SCP-2-OA-02 (black) complex, which attained a RMSD value of ~4.5–5 Å at 15 ns and remained in the same range until 45 ns. A slight drop was observed after 45 ns, but it gained the same range towards the end of simulation. The other two complexes, i.e., SCP-2-OA-06 and SCP-2-OA-09 showed more stable trajectories than the apo protein. The plot of SCP-2-OA-06 (red) showed that the RMSD remained in the range of ~3.5–4 Å until 35 ns and then a deviation of ~0.5 Å was observed from 35 to 45 ns, but it gained the same range towards the end of the simulation. The RMSD plot of SCP-2-OA-09 (green) showed a stable trend throughout the simulation. The stabilities of protein–ligand complexes were further validated by calculating the RMSD of ligand. It can be observed that the ligands showed stable plots with RMSD values ranging from ~0.5 to 1.5 Å except for OA-02 that showed a minor deviation at 30 ns, but it again attained the previous confirmations (Figure 2B). These plots showed the stability of protein–ligand complexes.

The dynamic behavior of the amino acid residues was determined by calculating the RMSF values. The higher RMSF values show the flexibility of residues or the loops of the protein, while the lower values describe the rigid part of protein, i.e., alpha helices and beta sheets. Higher RMSF values at N and C terminals can be observed as these are the loop regions (Figure 2C). A minor fluctuation was observed at residues 23 to 26 and 45 to 50 as these reside in loops. The remaining residues maintained their low flexibility, which showed the stability of protein and its complexes.

The protein structure stability was further analyzed by calculating the solvent accessible surface area. The distortion in the protein structure increased the total exposed area of protein accessible to solvent. So, if the SASA increases from the initial conformation, it shows a diffused protein structure, while low values elaborate the structure stability. The SASA values of the apo SCP-2 and its complexes are shown in Figure 2D. The SASA values of all systems were started from ~8500 Å2 and then dropped gradually to ~8000 Å2 at 10 ns. The SASA values remained in the range of ~7000 to 8000 Å2 until 30 ns and then minor deviations were observed in red and green plots, while black and blue remained stable. The overall SASA analysis showed that the protein structure remained stable in all systems and no diffusion was observed as the total accessible surface area to solvent was not increased. All these analyses demonstrated that the protein remained stable when bound to the OA-02, 06 and 09 compounds and no conformational changes were observed in protein structure.

The binding free energies of these complexes were also calculated by applying the MM/GBSA to the last 300 frames of MD trajectories. The total binding energy in terms of solvation energy, gas phase energy and entropic contributions were calculated. To find out the contribution of binding site residues, the systems were decomposed. The contribution of energy components, i.e., VDWAALS, EEL, EGB, ESURF, Delta G_gas_, Delta G_solv_ and total energy of each complex is given in Appendix A Appendix A. Moreover, the contribution of binding site residues was also calculated by energy decomposition function of MM/GBSA, which showed that GLN107, PHE110 and MET121 had high contribution in total binding free energy (Appendix A Appendix A).

### 3.4. Larval Mortality

When the 3rd instar larvae were exposed to different concentrations of OA-02, OA-06 and OA-09, a significantly different larval mortality, both due to type of compounds and their concentrations was observed (Kruskal–Wallis H = 6.33; *df* = 2; *p* = 0.042 and Kruskal–Wallis H = 9.30; *df* = 2; *p* = 0.026, respectively). Highest mortality of 3rd instar larvae was recorded when treated with 50 μM concentration of OA-09 (Figure 3A). Since the insects are capable of selective feeding [79], it can be assumed that avoiding excessive feeding on a diet with higher concentrations may have resulted in low mortalities as compared to those fed with a lower dosage. Whereas, the mortality of 4th and 5th instar larvae were not significantly affected by the type of compounds (Kruskal–Wallis H = 4.31; *df* = 2; *p* = 0.116 and Kruskal–Wallis H = 5.75; *df* = 2; *p* = 0.056, respectively) and their concentrations (Kruskal–Wallis H = 5.91; *df* = 2; *p* = 0.116 and Kruskal–Wallis H = 2.44; *df* = 2; *p* = 0.486, respectively) (Figure 3B,C). Similarly, based on the probit analysis, OA-02 was less toxic to H. armigera larvae reared on a treated artificial diet with highest values for LC50 (313.164 mg/L) (Table 3). On the other hand, OA-06 and OA-09 had LC50 values of 143.990 mg/L. OA-09 had the highest toxicity and the least LC50 value (60.956 mg/L) (Table 3).

### 3.5. Growth and Longevity of Different Stages

The data have demonstrated that the 3rd–5th instar larvae treated with different compounds have significantly different weight accumulations (*F* = 4.309; *df* = 2, 24; *p* = 0.025, *F* = 15.552; *df* = 2, 24; *p* < 0.001, *F* = 20.407; *df* = 2, 24; *p* < 0.001, respectively), while no significant difference due to variable concentrations was observed (Figure 4). The least weight accumulation was observed in the 3rd instar larvae treated with OA-02, while the highest weight accumulation was in the 5th instar larvae treated with the same compound (Figure 4). In contrast, the pupal weights appeared to be significantly affected due to variable concentrations of all compounds (*F* = 5.640; *df* = 3, 24; *p* = 0.005) (Figure 4) where pupae in control treatment accumulated the lower weight compared to those who were fed on treated diets during larval stages (Figure 4).

When the developmental periods of larvae, pupae and adults were observed, significant effects of compound used, and their concentrations were observed. Significantly shorter stadia lengths were observed (compound × concentration: *F* = 140.48; *df* = 6, 24; *p* < 0.001) when the larvae were fed on diet containing 450 μM concentration of OA-09 respect to other tested compounds (Table 4). When the pupal periods were observed, a significant difference due to interaction of compound type and their concentrations were noticed (compound × concentration: *F* = 11.137; *df* = 6, 24; *p* < 0.001) (Table 4). The larvae that were fed on OA-02 treated food (450 μM) ultimately spent the highest duration as pupae while all other treatments had similar pupal periods as that in the control (Table 4). In general, OA-02 and OA-06 had a similar effect on pupal period and 50 μM and 150 μM concentrations also had similar effects on pupal periods (Table 4).

Overall, when the total periods from 3rd instar larvae to adult emergence were observed, a significant difference due to interaction effect of the compounds and their concentrations were observed (compound × concentration: *F* = 4.044; *df* = 6, 24; *p* = 0.006) (Table 4). The trend was similar as observed in larval periods and stadia lengths. When emerged, the compounds and concentrations also had significant impact on longevity of both the male and female moths (*F* = 28.226; *df* = 6, 24; *p* < 0.001 and *F* = 19.399; *df* = 6, 24; *p* < 0.001, respectively). The shortest-lived male and female moths were recorded in the population that was reared on the 450 μM treated diet (Table 4). The adults from the population fed on OA-02 and OA-06 treated diet had statistically similar effects but significantly lower than control populations (Table 4).

### 3.6. Developmental and Reproductive Potentials

In terms of pupation (%) and adult emergence (%) significant effects of both the concentrations and compound types were observed (*F* = 28.926; *df* = 6, 24; *p* < 0.001 and *F* = 115.577; *df* = 6, 24; *p* < 0.001, respectively) (Table 5). All the different concentrations significantly reduced pupation and adult emergence rates as compared to the control populations. The OA-02 treated populations had significantly different and higher pupation and adult emergence rates as compared to OA-06 and OA-09 (Table 5).

As far as fecundity is concerned, we observed that all compounds at different concentrations had a significant effect on fecundity and fertility (compound × concentration: *F* = 7.376; *df* = 6, 24; *p* < 0.001 and *F* = 11.591; *df* = 6, 24; *p* < 0.001, respectively) (Table 5). The mean numbers of eggs laid did not differ significantly among the populations treated with all three concentrations but still was significantly lower as compared to the control. However, a significant effect on the egg hatch (fertility) was obvious where 450 μM of OA-09 reduced the fertility and only 58% of the eggs laid were hatched (Table 5).

### 3.7. Lifetable Parameters

The intrinsic (r) and finite (λ) rates of population increase were not affected by treating with different compounds, while in general, net reproductive rates (R_o_) and gross reproductive rate (GRR) were reduced as compared to the control population (Table 6). The results clearly demonstrated significantly reduced generation times among treated and control populations (Table 6). The population that was fed on OA-09 had the least net reproductive rate coupled with shortest generation time as compared to other compounds (Table 6).

Age-specific survival rates (*l*_x_) and fecundity (*m*_x_) of the populations that were fed on diets treated with OA-02, OA-06, OA-09 and control demonstrated a variable trend (Figure 5). The population feeding on OA-02 treated diets had survived better (*l*_x_ = 0.78) in the early larval stages and thereafter a quick population decline was observed until day 32 (Figure 5). The populations which were reared on OA-06 and OA-09 treated diet had declining survival rates in the early stages of development and then they became constant before a sharp fall at the 23rd day, respectively (Figure 5). In the control population, the decline in survival rates started at about the 37th day to the 40th day (Figure 5). When the fecundity (*m*_x_) was observed among these populations, a significant difference in the onset of oviposition was observed. Those reared on OA-09 diet started oviposition on the 21st day of their lifecycle, while the OA-06 reared population started egg laying on the 22nd day (Figure 5). The control and OA-02 reared populations had oviposited at the same time, i.e., the 24th day (Figure 5). Oviposition peaks in the population fed on OA-09 treated dyes were observed earlier, i.e., 21st day while the control population had the peak of oviposition observed around the 38th day of their lifecycle. The highest fecundity (*m*_x_) was observed in females that emerged from the OA-06 treated population and this might be due to a sublethal effect of the compound [80].

## 4. Discussion

There are many factors that affect herbivore population establishment on a particular host, availability of requisite nutrients is one of those significant factors. The variable chemical composition of the host plant impact their developmental, growth and survival [81]. Demographic evaluation [82] with either restricting the extraction of some key nutrients or blocking their metabolism after ingestion can provide an effective and ecofriendly pest control approach. In this study, we have evaluated the activity of some SCP-2 inhibitor compounds against *H. armigera* with the above-mentioned approach.

In previous studies, several candidate compounds from phytochemicals [83], natural products [84] and commercial databases [85] have been identified by in silico studies. So, by applying the computer aided drug design approaches, we identified three compounds from a series of quinolone azines synthetic compounds. The binding poses of the docked compounds were assessed and the compounds having similar binding mode to cholesterol, a natural substrate of Ha-SCP2, were selected for the protein–ligand stability analysis by MD Simulation. The RMSD of protein backbone atoms helps to study the stability of protein [86] when a certain ligand is bound to it. The RMSD values of our studied complexes showed stable confirmations with minor deviations (Figure 2A,B). Similarly, the residual flexibility of protein was estimated by RMSF analysis [87], which indicated the stable protein structure in all three complexes (Figure 2C). To find the binding free energy in the selected complexes, a MM/GBSA module was applied on the last 300 snapshots of MD trajectories. MM/GBSA provides accurate estimates of binding affinity in terms of binding free energy [88]. The total binding free energy values −24.42 ± 0.33, −21.35 ± 0.45, −28.19 ± 0.40 for OA-02, OA-06 and OA-09, respectively, showed that the compounds were bound with protein with reasonably good affinity. Moreover, the contribution of binding site residues in total binding free energy was estimated (Appendix A) and based on the contributions, key residues Gln107, Phe110 and Met121, were identified. Further, the three selected compounds were used for in vivo evaluation.

In general, our results have demonstrated that the growth patterns and population increase are significantly affected in the populations when treated with different doses of SCP-2 inhibiting molecules. These findings have important implications in developing a targeted and environmentally safe management strategy against *H. armigera*. Although the higher concentrations did not kill the larvae when fed on it, the life table parameters and developmental variables were greatly affected, resulting in decreased larval and pupal weights, decreased pupation and eclosion, as well as decreased fertility (Table 3 and Table 4). These findings are in accordance with earlier studies reporting a significant decrease in pupal weights of *H. armigera* [55,89] and fertility [90] when reared on different insecticidal admixed artificial diets.

Our findings have demonstrated that, contrary to other studies depicting the dose dependent effect on insect mortality [91,92], lower mortality was observed when the larvae were treated with the higher concentrations of the test compounds. This might be due to the nature of the compound, as the test molecules in our study were SCP-2 inhibitors that may act as insect growth regulators (IGR) rather than as a toxin [91,92,93].

We also noticed that fecundity was not significantly affected, but the fertility was greatly reduced in the populations where the larvae were fed on higher concentrations of the SCP-2 inhibitors. This clearly suggests that the compounds may have decreased palatability and cholesterol accumulation and transportation in larval bodies ultimately affecting reproductive potentials [94,95]. The second major impact of the treated compounds was observed in the form of reduced stadia lengths as even it varies when *H. armigera* feed on different plant species [96,97]. The larval period was shortened when treated with test compounds; however, the pupal periods remained unaffected. We assume that disruption in ample accumulation of cholesterol and sterol that is required in chitin synthesis [90,98] resulted in the larvae molting earlier than control. Clearly, OA-09 at the rate of 450 μM had demonstrated promising results where the larval periods, weights, pupal periods, weights and ultimately the female longevity and fertility were negatively affected in the test populations. Such decrement in significant life parameters of *H. armigera* can be attributed to reduced feeding efficiency and disruption of sterol carrier protein functions [53].

## 5. Conclusions

In conclusion, the life table parameters of *H. armigera* were significantly reduced when treated with OA-09. The reduced larval and pupal durations and oviposition of females may reduce population and growth. Although the generation time is also reduced due to test compounds, coupled with significantly reduced female longevity and fertility (up to 42%), it suggests that the pest establishment will be seriously compromised in terms of population growth if these novel compounds are applied. This conclusion can further be supported by at least 70% decreased net reproductive rates. Our findings have significant implications for environmentally sustainable pest control strategies by reducing the dependence on traditional chemical insecticide usage.

## Figures and Tables

**Figure 1 insects-13-01169-f001:**
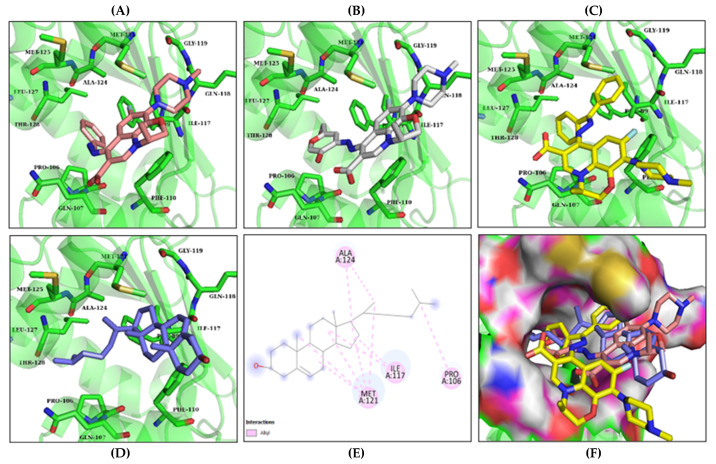
Predicted binding modes of OA-02, OA-06 and OA-09. (**A**–**D**) Sticks model representation of OA-02 (Brown), OA-06 (White), OA-09 (Yellow) and Cholesterol (Blue) binding with active site. (**E**) Interactions of Cholesterol with the binding pocket. (**F**) Representation of binding modes of selected compounds aligned on cholesterol in the binding pocket showing in surface.

**Figure 2 insects-13-01169-f002:**
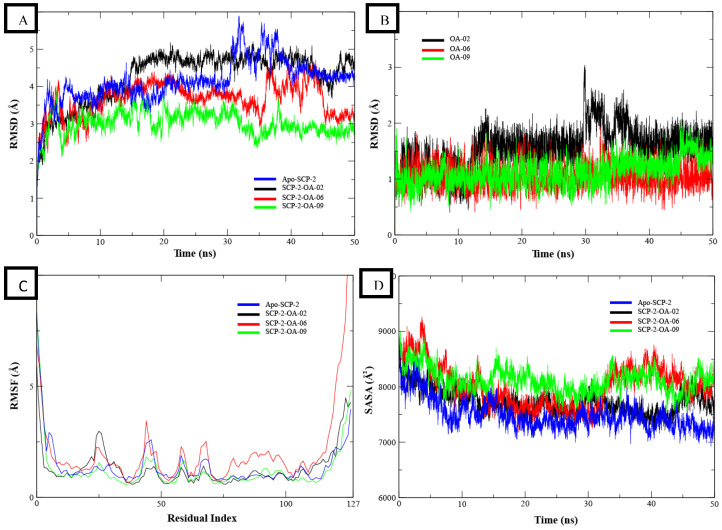
The binding stability analysis of protein–ligand complexes by MD Simulation. (**A**) RMSD plots of backbone of protein (blue) and its complexes OA-02 (black), OA-06 (red) and OA-09 (green). (**B**) The RMSD plots of ligands. (**C**) The residual flexibility analysis of the protein and complexes. (**D**) The solvent exposed area analysis of protein and its complexes.

**Figure 3 insects-13-01169-f003:**
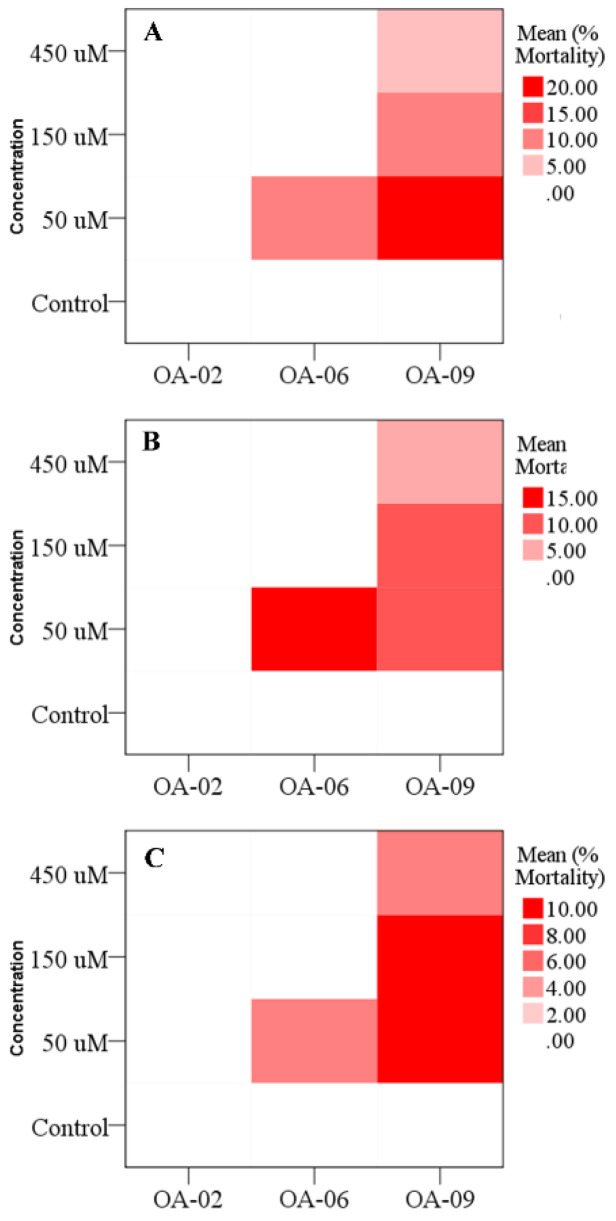
Mean mortality percentages of (**A**) 3rd instar (Compound: H = 6.33; *df* = 2; *p* = 0.042 and Concentration: H = 9.30; *df* = 2; *p* = 0.026), (**B**) 4th instar (Compound: H = 4.31; *df* = 2; *p* = 0.116 and Concentration: H = 5.91; *df* = 2; *p* = 0.116) and (**C**) 5th instar larvae (Compound: H = 3.75; *df* = 2; *p* = 0.056 and Concentration: H = 2.114; *df* = 2; *p* = 0.486) of Helicoverpa armigera when treated with different concentrations of OA-02, OA-06 and OA-09 compounds. The data were subjected to nonparametric Kruskal–Wallis H test.

**Figure 4 insects-13-01169-f004:**
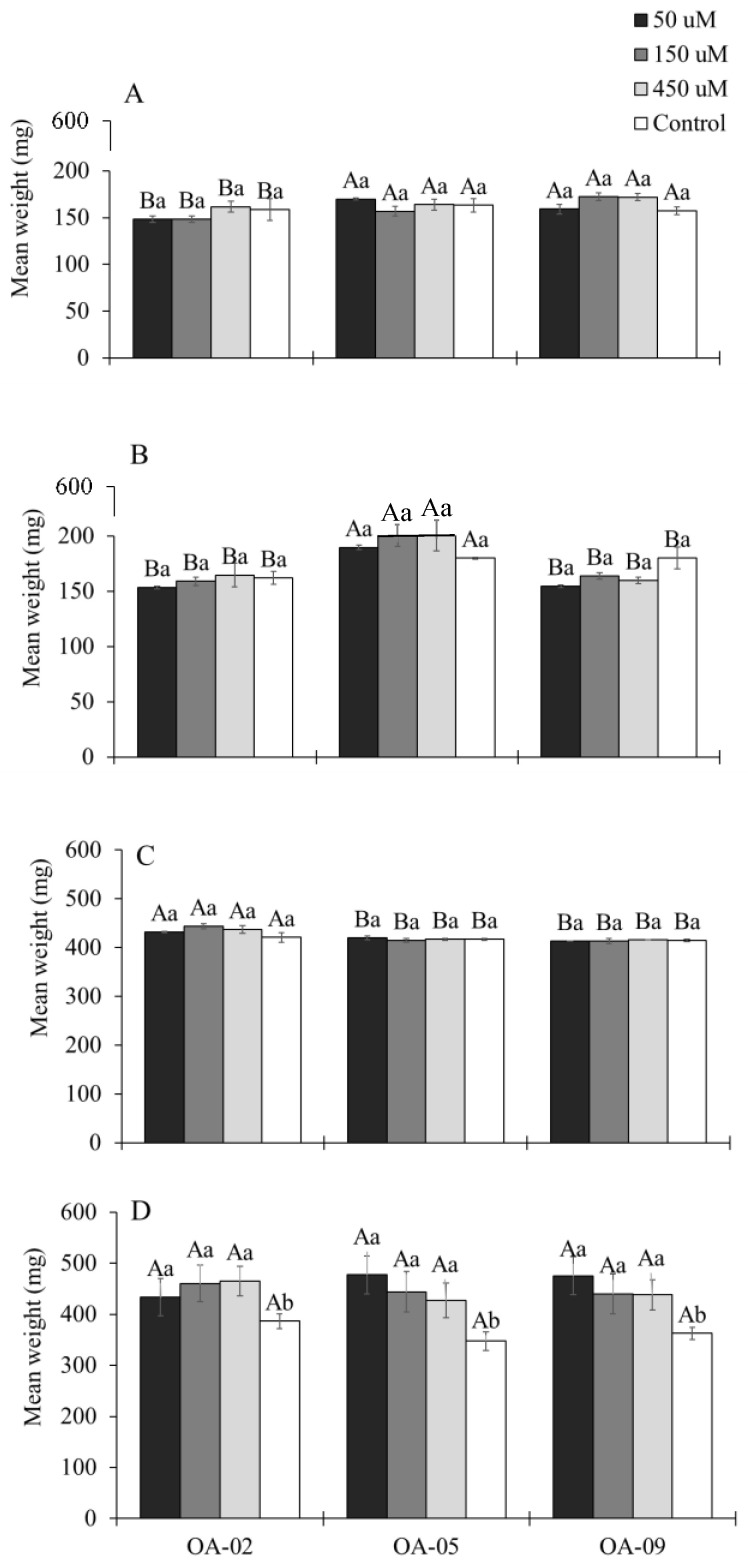
Effect of different concentrations of OA-02, OA-06 and OA-09 compounds on 3rd–5th instar H. armigera larval (**A**–**C**) and pupal weights (**D**). The data were subjected to a 2-way ANOVA (statistics for the effects of compound type for 3rd–5th instar larval weight and pupal weight: F_2, 24_ = 4.301; *p* = 0.025, *F*_2, 24_ 15.552; *p* < 0.001, *F*_2, 24_ = 20.407; *p* < 0.001 and *F*_2, 24_ = 0.156; *p* = 0.856, respectively while the statistics for the effect of their concentrations: *F*_2, 24_ = 1.003; *p* = 0.408, *F*_2, 24_ = 0.904; *p* = 0.454, *F*_2, 24_ = 1.083; *p* = 0.375 and *F*_2, 24_ = 5.640; *p* = 0.005, respectively). The bars represent mean ± SEM and the superscript uppercase and lowercase letters atop each bar represent the post-hoc pairwise comparisons between different compounds and concentrations, respectively. The bars with different letters are significantly different from each other (*p* < 0.05).

**Figure 5 insects-13-01169-f005:**
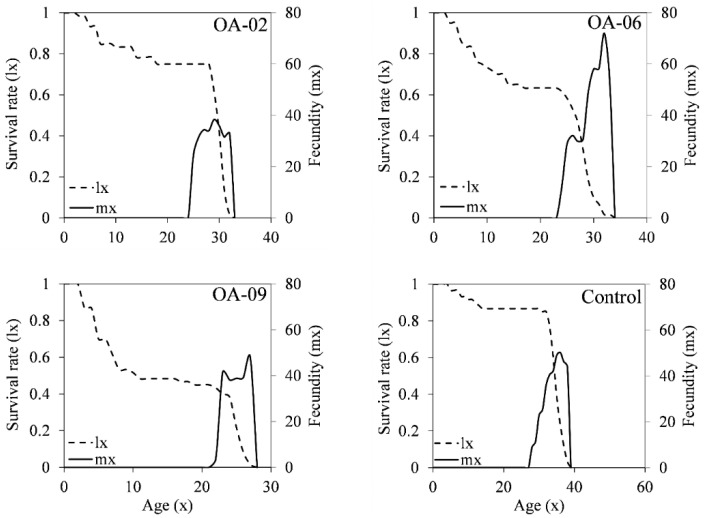
Age-specific survival (*l*_x_) and age-specific fecundity (*m*_x_) of Helicoverpa armigera when treated with OA-02, OA-06 and OA-09 compounds.

**Table 1 insects-13-01169-t001:** Predicted toxicity properties of screened SCP-2 inhibitors.

ID	Mol. Wt.	log *P*	Tumorigenicity	Reproductive Effect	Irritant	Mutagenicity
OA-01	457.50	1.31	None	High	High	High
OA-02	463.51	2.85	None	None	None	None
OA-03	509.51	2.93	None	None	None	None
OA-04	542.40	3.57	None	None	None	None
OA-05	479.51	2.50	None	None	None	None
OA-06	499.49	0.14	None	None	None	None
OA-07	509.53	2.43	Low	None	None	None
OA-08	523.56	2.71	None	None	None	None
OA-09	539.60	4.04	None	None	None	None

**Table 2 insects-13-01169-t002:** The glide scores of the OA series compounds with corresponding structures.

***OA-01*** 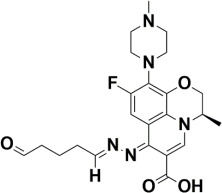 −5.341	***OA-02*** 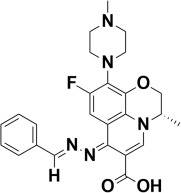 −5.369	***OA-03*** 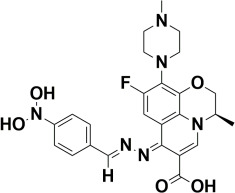 −7.666
***OA-04*** 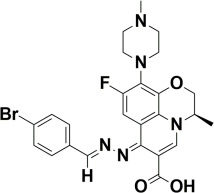 −4.505	***OA-05*** 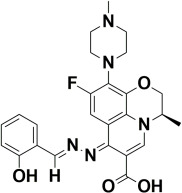 −4.254	***OA-06*** 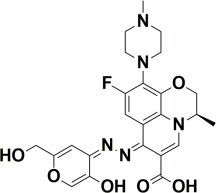 −3.365
***OA-07*** 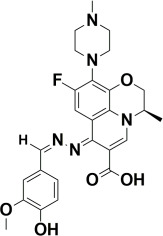 −4.305	***OA-08*** 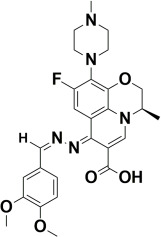 −4.587	***OA-09*** 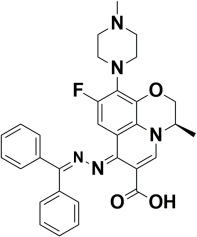 −5.031

**Table 3 insects-13-01169-t003:** LC50 and LC90 values of OA-02, OA-06 and OA-09 compounds against H. armigera larvae along with their probit model fit parameters.

Parameters	OA-02	OA-06	OA-09
n	60	60	60
LC_50_ (95% FL) (mg/L)	313.2 (194.2–845.5)	143.9 (80.6–250.6)	60.9 (21.1–98.6)
LC_90_ (95% FL) (mg/L)	7248.7 (1843.4–532,962.6)	3778.2 (1141.1–175,820.4)	1235.9 (533.5–14,300)
Probit model fit			
χ^2^	0.90	1.20	1.80
df	1	1	1
Slope ± SE	0.93 ± 0.25	0.91 ± 0.25	1.01 ± 0.26
*p*	0.336	0.183	0.279

**Table 4 insects-13-01169-t004:** Effect of different concentrations of OA-02, OA-06 and OA-09 compounds on developmental periods during different life stages of Helicoverpa armigera.

Parameters (Days)	Compound	Concentrations				Test Statistics
		50 μM	150 μM	450 μM	Control	
Larval duration	OA-02	13.36 ± 0.05 ^Aa^	13.42 ± 0.07 ^Ab^	5.76 ± 0.07 ^Ac^	15.03 ± 0.53 ^Aa^	*F*_2, 24_ = 140.48 *p* < 0.001
	OA-06	16.14 ± 0.12 ^Ba^	7.14 ± 0.27 ^Bb^	3.93 ± 0.01 ^Bc^	11.38 ± 2.25 ^Ba^	
	OA-09	10.00 ± 0.10 ^Ca^	6.01 ± 0.04 ^Cb^	3.75 ± 0.02 ^Cc^	11.25 ± 2.16 ^Ca^	
Pupal duration	OA-02	16.19 ± 0.12 ^Aa^	15.41 ± 0.11 ^Aa^	17.12 ± 0.03 ^Aa^	15.55 ± 0.74 ^Aa^	*F*_2, 24_ = 11.137 *p* < 0.001
	OA-06	14.41 ± 0.15 ^Ba^	13.39 ± 0.03 ^Ba^	14.42 ± 0.09 ^Ba^	15.53 ± 0.83 ^Ba^	
	OA-09	10.96 ± 0.30 ^Ca^	12.23 ± 0.06 ^Ca^	12.95 ± 0.02 ^Ca^	15.16 ± 0.87 ^Ca^	
3rd Instar to adult emergence	OA-02	30.56 ± 0.16 ^Aa^	29.84 ± 0.18 ^Ab^	23.88 ± 0.10 ^Ac^	28.25 ± 3.25 ^Aa^	*F*_2, 24_ = 4.044 *p* = 0.006
	OA-06	31.55 ± 0.26 ^Ba^	21.53 ± 0.24 ^Bb^	19.36 ± 0.11 ^Bc^	27.91 ± 2.77 ^Ba^	
	OA-09	21.96 ± 0.36 ^Ca^	19.25 ± 0.10 ^Cb^	17.70 ± 0.04 ^Cc^	27.41 ± 2.71 ^Ca^	
Longevity of male moths	OA-02	8.47 ± 0.18 ^Ab^	7.79 ± 0.15 ^Ab^	7.14 ± 0.09 ^Ac^	9.01 ± 0.05 ^Aa^	*F*_2, 24_ = 28.226 *p* < 0.001
	OA-06	6.03 ± 0.04 ^Bb^	5.70 ± 0.05 ^Bb^	5.32 ± 0.16 ^Bc^	8.19 ± 0.03 ^Ba^	
	OA-09	4.62 ± 0.06 ^Cb^	5.10 ± 0.10 ^Cb^	5.40 ± 0.20 ^Cc^	8.06 ± 0.12 ^Ca^	
Longevity of female moths	OA-02	8.92 ± 0.03 ^Ab^	8.68 ± 0.11 ^Ab^	8.23 ± 0.13 ^Ac^	10.02 ± 0.05 ^Aa^	*F*_2, 24_ = 19.399 *p* < 0.001
	OA-06	7.41 ± 0.09 ^Bb^	7.07 ± 0.08 ^Bb^	6.26 ± 0.13 ^Bc^	9.34 ± 0.08 ^Ba^	
	OA-09	6.03 ± 0.08 ^Cb^	6.54 ± 0.02 ^Cb^	5.80 ± 0.15 ^Cc^	9.10 ± 0.15 ^Ca^	

The data are presented as mean ± SEM and the superscript uppercase and lowercase letters represent the post-hoc pairwise comparisons between compounds and the concentrations, respectively. The data with different letters in superscript differ significantly from each other (*p* < 0.05, Tukey’s HSD).

**Table 5 insects-13-01169-t005:** Developmental and reproductive parameters of Helicoverpa armigera when treated with different concentrations of OA-02, OA-06 and OA-09 compounds.

Parameters		Concentrations				Test Statistics
		50 μM	150 μM	450 μM	Control	
Pupation (%)	OA-02	88.00 ± 1.15 ^Ab^	84.39 ± 0.58 ^Ab^	81.34 ± 0.49 ^Ab^	95.00 ± 0.57 ^Aa^	*F*_6,24_ = 28.926 *p* < 0.001
	OA-06	74.93 ± 0.19 ^Bb^	80.41 ± 0.59 ^Bb^	80.86 ± 0.70 ^Bb^	97.66 ± 1.45 ^Ba^	
	OA-09	79.78 ± 0.28 ^Bb^	76.33 ± 0.88 ^Bb^	82.96 ± 0.35 ^Bb^	94.93 ± 0.17 ^Ba^	
Adult emergence (%)	OA-02	84.29 ± 0.41 ^Ab^	81.40 ± 0.43 ^Ac^	81.46 ± 1.08 ^Ad^	94.35 ± 0.54 ^Aa^	*F*_6,24_ = 115.577 *p* < 0.001
	OA-06	79.96 ± 0.20 ^Bb^	80.61 ± 0.77 ^Bc^	74.70 ± 0.35 ^Bd^	86.33 ± 0.88 ^Ba^	
	OA-09	61.95 ± 0.21 ^Cb^	54.75 ± 1.14 ^Cc^	54.94 ± 0.33 ^Cd^	90.25 ± 0.49 ^Ca^	
Fecundity (numbers)	OA-02	415.00 ± 2.88 ^Ab^	461.66 ± 4.40 ^Ab^	405.33 ± 7.12 ^Ab^	513.71 ± 3.60 ^Aa^	*F*_6,24_ = 7.376 *p* < 0.001
	OA-06	391.80 ± 1.74 ^Bb^	394.72 ± 2.84 ^Bb^	409.09 ± 2.61 ^Bb^	499.31 ± 25.53 ^Ba^	
	OA-09	305.66 ± 3.84 ^Cb^	291.33 ± 1.45 ^Cb^	279.44 ± 5.05 ^Cb^	448.31 ± 15.64 ^Ca^	
Fertility (%)	OA-02	84.35 ± 1.74 ^Ab^	73.64 ± 0.95 ^Ac^	74.83 ± 0.23 ^Ad^	87.48 ± 1.61 ^Aa^	*F*_6,24_ = 11.591 *p* < 0.001
	OA-06	72.73 ± 0.41 ^Bb^	64.69 ± 0.21 ^Bc^	61.02 ± 0.23 ^Bd^	82.88 ± 1.88 ^Ba^	
	OA-09	64.67 ± 0.45 ^Cb^	64.01 ± 0.73 ^Cc^	58.82 ± 0.22 ^Cd^	82.48 ± 0.99 ^Ca^	

The data are presented as mean ± SEM and the superscript uppercase and lowercase letters atop each bar represent the post-hoc pairwise comparisons between compounds and the concentrations, respectively. The data with different letters in superscript differ significantly from each other (*p* < 0.05, Tukey’s HSD).

**Table 6 insects-13-01169-t006:** Lifetable parameters of Helicoverpa armigera populations exposed to OA-02, OA-06 and OA-09 compounds.

Lifetable Indices	Compounds				Test Statistics
	OA-02	OA-06	OA-09	Control	
Intrinsic rate of increase (r)	0.17 ± 0.01 ^a^	0.16 ± 0.01 ^a^	0.17 ± 0.01 ^a^	0.16 ± 0 ^a^	*F*_3, 399996_ = 86675.405 *p* < 0.001
Finite rate of increase (λ)	1.19 ± 0.01 ^a^	1.17 ± 0.01 ^a^	1.18 ± 0.01 ^a^	1.17 ± 0.01 ^a^	*F*_3, 399996_ = 87349.024 *p* < 0.001
Net reproductive rate (R_o_)	136.43 ± 22.88 ^b^	86.27 ± 17.12 ^c^	57.0 ± 13.64 ^c^	183.48 ± 26.49 ^a^	*F*_3, 399996_ = 730371.962 *p* < 0.001
Gross reproductive rate (GRR)	262.86 ± 37.93 ^b^	424.82 ± 75.65 ^a^	236.36 ± 43.84 ^b^	376.86 ± 32.79 ^ab^	*F*_3, 399996_ = 262973.719 *p* < 0.001
Mean generation time (T)	28.32 ± 0.1 ^b^	27.67 ± 0.38 ^b^	24.43 ± 0.17 ^c^	33.14 ± 0.27 ^a^	*F*_3, 399996_ = 20563756.674 *p* < 0.001

The data presented are mean ± SEM while the superscript lowercase letters represent the post-hoc pairwise comparisons between compounds. The data with different letters in superscript differ significantly from each other (*p* < 0.05, Tukey’s HSD). The above data were bootstrapped (100,000 times) to calculate variances and then analyzed using a one-way ANOVA.

## Data Availability

All data and materials are available after publication for academic use only.

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
