# Peer review of "In Silico and In Vivo Evaluation of Synthesized SCP-2 Inhibiting Compounds on Life Table Parameters of *Helicoverpa armigera* (Hübner)"

_insects, 2022, doi:10.3390/insects13121169_

Round 1

Reviewer 1 Report

The content is of interest for the control of H. armigera, but several important aspects must be reviewed and adjusted before the work can be published. Some of them are mentioned below: 

Regarding the analysis: Since larval mortality is a response variable subject to the same factors (compound and concentration), Kruskal-Wallis is not an optimal tool for the analysis, because it is a one-way analysis. In fact here you do not specify either the factor considered in the K-W analysis (Compounds? or concentrations of each compound considered apart?)

You should consider two options for the analysis of mortality in a 2-way factorial arrangement: 

1-Generalized linear model assuming a binomial distribution (since proportions are derived from binomial data live/death). 

2-Transforming proportions with arcsine square root transformation and make the 2-way factorial ANOVA with transformed data.  

Figure 3: If the proportion of mortality is going to be presented considering the effect of the two factors (compounds and concentrations), it is more appropriate to use another type of graph that allows visualizing the standard error of the proportion for each treatment (i.e. each compound*concentration combination)

Description of results in some cases is not consistent with tables or figures presented, such as: 

Line 374: According to table 4, larval duration in OA-09  was the highest, not the lowest, and clearly lower than the observed in the control. 

Table 6 should include the effect of compounds and concentrations, which are discussed in the text, but are not presented.

Figure 5: You should note and discuss that fecundity was highest in OA-06 and why

Line 477:  "Our findings have demonstrated that the higher concentrations of test compounds had  reduced mortality" This is not observed in the tables or figures presented. Neither the LC50 or LC90 values, nor the survival curves support this statement.

line 487:" The larval period was shortened and hence the pupal periods increased significantly due to reduced accumulation of cholesterol and sterol that is  required in chitin synthesis"

 It is true that the larval period was shortened, but according to Table 4, the duration of pupa remained the same as in the control or was even shortened in several cases. In no case was the duration of pupa significantly greater than in the control. Additionally, it is usual that the duration of states lengthens as a consequence of the effect of compounds that negatively affect development. Your results show the opposite and you should discuss why this happens.

Additional important comments can be found in the attached file

Reviewer 2 Report

In this study, in house synthesized small molecules, with a tendency to disrupt insect molting, were evaluated against a Helicoverpa armigera. Based on in silico studies, three compounds i.e., OA-02, OA-06, and OA-09 were selected because they showed the same binding pose as the natural substrate cholesterol. The hydrophobic moieties of selected compounds interacted with the same residues and reside in the same position in the binding pocket. These three compounds were evaluated in vivo. One of the tested compounds significantly reduced larval and pupal weight accumulations and prolonged stadia lengths resulting into disrupted population growth. Furthermore, the emerged females had reduced fertility. The study is well conducted and results are scientifically sound. Some minor corrections are needed.

- Please review the formulas and the terms in each as there are parameters explained in the text that are missing from the formulas.

-Page 6 L 240.  “…the means were analyses with ANOVA followed …“ analyzed?

-Figure 1. Letters A-F are missing from the panels

- In figure 3 it is striking that there is no dose-response in larval mortality in each larval stage. The higher the concentration, the lower the mortality. Can you explain this result?

-Page 11 Line 358-359. The text contradicts the graphic. According to figure 4D, the control had the lowest pupal weight.

-figure 4. Legend: there is no asterisk in the figure.  

-Page 12 Line 368. Both type of compounds?

-Page 15 Line 427. “The data presented is mean ± SEM while the superscript uppercase and lowercase letters atop each bar represent the post-hoc pair-wise comparisons between compounds and the concentrations, respectively.” There are no concentration results. There are no uppercase letters in the table 6.

Round 2

Reviewer 1 Report

The authors satisfactorily incorporated the corrections and provided a sufficient explanation in some cases that they did not consider appropriate to make changes. 

Some minor corrections are included in the attached document

Authors should consider including the following references

Previous relevant works should be cited: 

TY  - JOUR

AU  - Haihao, ma

AU  - Ma, Yuemin

AU  - Liu, Xuehui

AU  - Dyer, David

AU  - Xu, Pingyong

AU  - Liu, Kaiyu

AU  - Lan, Q.

AU  - Hong, Huazhu

AU  - Peng, Jianxin

AU  - Peng, Rong

PY  - 2015/12/10

SP  - 18186

T1  - NMR structure and function of Helicoverpa armigera sterol carrier protein-2, an important insecticidal target from the cotton bollworm

VL  - 5

DO  - 10.1038/srep18186

JO  - Scientific Reports

ER  - 

________________________________

Xin Du, Haihao Ma, Xin Zhang, Kaiyu Liu, Jianxin Peng, Que Lan, Huazu Hong,

Characterization of the sterol carrier protein-x/sterol carrier protein-2 gene in the cotton bollworm, Helicoverpa armigera,

Journal of Insect Physiology,

Volume 58, Issue 11,

2012,

Pages 1413-1423,

ISSN 0022-1910,

https://doi.org/10.1016/j.jinsphys.2012.08.005.

(https://www.sciencedirect.com/science/article/pii/S0022191012002065)

Abstract: Cholesterol is a membrane component and the precursor of ecdysteroids in insects, but insects cannot synthesize cholesterol de novo. Therefore, cholesterol uptake and transportation during the feeding larval stages are critical processes in insects. The sterol carrier protein-2 domain (SCP-2) in sterol carrier proteins-x (SCP-x) has been speculated to be involved in intracellular cholesterol transfer and metabolism in vertebrates. However, a direct association between SCP-x gene expression, cholesterol absorption and development in lepidopteran insects is poorly understood. We identified the Helicoverpa armigera sterol carrier protein-x/2 (HaSCP-x/2) gene from the larval midgut cDNAs. The HaSCP-x/2 gene is well conserved during evolution and relatively divergent in heterogenetic species. Transcripts of HaSCP-x/2 were detected by qRT-PCR at the highest level in the midgut of H. armigera during the larval stages. Expression knockdown of HaSCP-x/2 transcripts via dsRNA interference resulted in delayed larval development and decreased adult fecundity. Sterol carrier protein-2 inhibitors were lethal to young larvae and decreased fertility in adults emerged from treated elder larvae in H. armigera. The results taken together suggest that HaSCPx/2 gene is important for normal development and fertility in H. armigera.

Keywords: Sterol carrier protein-2; Cholesterol; Development; Fertility; RNA interference

__________________________________

RT Journal Article

A1 Cai, Jun

A1 Du, Xinkai

A1 Wang, Changgao

A1 Lin, Jianguo

A1 Du, Xin

T1 Identification of Potential Helicoverpa armigera (Lepidoptera: Noctuidae) Sterol Carrier Protein-2 Inhibitors Through High-Throughput Virtual Screening

JF Journal of Economic Entomology

JO J Econ Entomol

YR 2017

DO 10.1093/jee/tox157

VO 110

IS 4

SP 1779

OP 1784

SN 0022-0493

AB Helicoverpa armigera sterol carrier protein-2 (HaSCP-2) is a validated target for development of novel insecticides due to its divergent protein structure and function from the vertebrate SCP-2. HaSCP-2 is important for normal development and fertility in Helicoverpa armigera (Hübner). The discovery of chemical inhibitors of HaSCP-2 through a structure-based virtual screening is reported here. Bioassay indicated that H1 and H14 had inhibitory effect on the growth of H. armigera larvae. The results suggest that H1 and H14 are promising as useful lead compounds for further optimization and development of novel SCP-2-specific pesticides.

RD 11/4/2022

UL https://doi.org/10.1093/jee/tox157
